# Evolutionary dynamics in the Anthropocene: Life history and intensity of human contact shape antipredator responses

Benjamin Geffroy[1]⊙*, Bastien Sadoul[1]⊙, Breanna J. Putman[2,3]¤, Oded Berger-Tal[4], László Zsolt Garamszegi[5,6], Anders Pape Møller[7,8], Daniel T. Blumstein[2]

**1** MARBEC, Univ Montpellier, Ifremer, IRD, CNRS, Palavas-Les-Flots, France, **2** Department of Ecology and Evolutionary Biology, University of California, Los Angeles, California, United States of America, **3** Natural History Museum of Los Angeles County, Department of Herpetology and Urban Nature Research Center, Los Angeles, California, United States of America, **4** Mitrani Department of Desert Ecology, Jacob Blaustein Institutes for Desert Research, Ben-Gurion University of the Negev, Israel, **5** Centre for Ecological Research, Institute of Ecology and Botany, Vácrátót, Hungary, **6** MTA-ELTE, Theoretical Biology and Evolutionary Ecology Research Group, Department of Plant Systematics, Ecology and Theoretical Biology, Eötvös Loránd University, Budapest, Hungary, **7** Laboratoire d'Ecologie, Systematique et Evolution, Centre National de la Recherche Scientifique, Universite Paris-Sud, France, **8** Ministry of Education Key Laboratory for Biodiversity Science and Ecological Engineering, College of Life Sciences, Beijing Normal University, Beijing, China

⊙ These authors contributed equally to this work.
¤ Current address: Department of Biology, California State University San Bernardino, San Bernardino, California, United States of America
* bgeffroy@ifremer.fr

**Data Availability Statement:** All relevant data are within the paper and its Supporting Information files.

## Abstract

Humans profoundly impact landscapes, ecosystems, and animal behavior. In many cases, animals living near humans become tolerant of them and reduce antipredator responses. Yet, we still lack an understanding of the underlying evolutionary dynamics behind these shifts in traits that affect animal survival. Here, we used a phylogenetic meta-analysis to determine how the mean and variability in antipredator responses change as a function of the number of generations spent in contact with humans under 3 different contexts: urbanization, captivity, and domestication. We found that any contact with humans leads to a rapid reduction in mean antipredator responses as expected. Notably, the variance among individuals over time observed a short-term increase followed by a gradual decrease, significant for domesticated animals. This implies that intense human contact immediately releases animals from predation pressure and then imposes strong anthropogenic selection on traits. In addition, our results reveal that the loss of antipredator traits due to urbanization is similar to that of domestication but occurs 3 times more slowly. Furthermore, the rapid disappearance of antipredator traits was associated with 2 main life-history traits: foraging guild and whether the species was solitary or gregarious (i.e., group-living). For domesticated animals, this decrease in antipredator behavior was stronger for herbivores than for omnivores or carnivores and for solitary than for gregarious species. By contrast, the decrease in antipredator traits was stronger for gregarious, urbanized species, although this result is based mostly on birds. Our study offers 2 major insights on evolution in the Anthropocene: (1) changes in traits occur rapidly even under unintentional human "interventions" (i.e., urbanization) and (2) there are similarities between the selection pressures exerted by

**Funding:** LZG is supported by the National Research, the Development and Innovation Office in Hungary (K129215). DTB is supported by the National Science Foundation and the Australian Research Council. But the funders had no role in study design, data collection and analysis, decision to publish, or preparation of the manuscript.

**Competing interests:** "The authors have declared that no competing interests exist."

**Abbreviations:** CI, confidence interval; cma-es, covariance matrix adapting evolutionary strategy; CV, coefficient of variation; DIC, Deviance Information Criterion; HIREC, Human-Induced Rapid Environmental Change; MAE, mean absolute error; MCMCglmm, Markov chain Monte Carlo generalized linear mixed model.

domestication and by urbanization. In all, such changes could affect animal survival in a predator-rich world, but through understanding evolutionary dynamics, we can better predict when and how exposure to humans modify these fitness-related traits.

## Introduction

Over the last few centuries, humans have transformed landscapes and ecosystems, with a relatively sudden transition to a highly modified world [1]. This rapid transition has created a major biodiversity crisis [2,3] and is threatening as many as 1 million species with extinction [4]. Animals have been particularly impacted by different anthropogenic disturbances, termed Human-Induced Rapid Environmental Change (HIREC) [5]. These include extensive harvesting, pollution, habitat loss and fragmentation, tourism, urbanization, and climate change [5]. While in many cases, these disturbances directly lead to species decline, they can also elicit more nuanced changes to fitness-related traits [6], including antipredator responses, which evolved in natural habitats but may become less important when animals are in contact with humans.

Prior to human contact, many species were likely under strong selection to avoid predation [7]. Nevertheless, mounting an antipredator response is costly in terms of energy since less time is available for other activities such as foraging and reproduction [8,9]. This trade-off is context-dependent, with the allocation of antipredator responses increasing with predation risk [10]. In nature, variability in antipredator behavior depends on both internal and external factors. Internal factors include the state of the individual (for example, how hungry it is), its motivations, and both inter- and intraindividual differences (for example, personality type) [11–13], while external factors include predator density, abundance, and hunting style [14,15].

Human disturbance, whether passively (by merely being present) or actively (through interventions to animal populations), inherently changes predation risk and the trade-offs associated with antipredator responses. Humans may create a human shield whereby association with humans or human infrastructure protects prey species from predation [16], and consequently, anthropogenic habitats may be characterized as having lower predation risk than more natural areas [17]. In response, animals may alter their antipredator traits by means of behavioral flexibility (i.e., the animal learns to modify its behavior as a response to a changing environment), developmental plasticity (i.e., the change in behavior is triggered by the developmental conditions experienced by the individual), rapid evolutionary adaptation, or a combination of any of these mechanisms [18].

To date, we have yet to understand the traits and conditions that allow certain species to successfully adapt to HIREC while others are unable to do so [19]. Recent evidence suggests that differences in evolutionary responses could be driven by within-population variance in traits [20], emphasizing the importance of behavioral and life-history traits in predicting the response of organisms to environmental change. Indeed, species (and populations) may differ in a variety of life-history traits, and these traits may influence the rate and magnitude of change in antipredator responses in anthropogenic environments. For example, it has been shown that species with a slower pace of life are more prone to change their antipredator behavior in the context of both urbanization [21] and domestication [22]. Other traits may be closely connected with flexibility in responses. For instance, sociality (i.e., group living) or foraging behavior are both central in defining the life history of a species (or population) and consequently may shape how species cope with environmental change. To better predict which species are vulnerable to HIREC, we must understand how the context of human contact,

along with life-history traits, shape the evolutionary dynamics of fitness-related traits such as antipredator responses. Nonetheless, it is important to remember that the responses of animals to anthropogenic change are often maladaptive and may create evolutionary and ecological traps [23,24] or be insufficient to overcome novel selection pressures [25].

For domesticated and captive animals, the threat of predation is substantially reduced or even eliminated (although we recognize that some domesticated animals, such as free-ranging livestock, may still face threats of predation [26]). During urbanization, human influence is assumed to be less drastic, but multiple studies have shown that urban areas offer relaxed predation pressure for some prey species (and consequently, many of these species flourish in urban environments) [27–29]. Because all 3 contexts may relax predation pressure, we assume that all may drive evolutionary shifts in antipredator traits by modifying both the traits' means and interindividual variances, as detected in the wild (for example, [28]), urbanized (for example, [29]), captive (for example, [30]), and domesticated (for example, [31]) contexts. While it is clear that relaxed predation rate is expected to reduce the overall means of antipredator traits, we lack knowledge on how rapidly these traits are lost and to what extent this depends on human-contact intensity in conjunction with species life history. In addition, looking at how the variances of these traits change with time can give us a unique and novel glimpse into the mechanisms through which animals respond and presumably adapt to reduced predation risk in different human-made and natural contexts alike.

We therefore conducted a comprehensive phylogenetic meta-analysis to investigate the dynamics of antipredator responses under this relaxed-selection hypothesis in 3 different contexts: urbanization, captivity, and domestication. We tested the hypothesis that the gradient of human influence from low to high (urbanization < captivity < domestication) is directly linked to the speed at which (1) the mean value of antipredator traits decreases and (2) the variation of antipredator traits increases (which would indicate relaxed selection on traits). Moreover, we tested how different life-history traits may influence the changes in antipredator responses seen in different species.

## Results

All 3 types of human influences had the same pattern of mean response: a rapid reduction in antipredator phenotypes (Fig 1A and Fig 1B, S1 Table). The first derivatives of these curves show that domestication is the context with the most rapid change in antipredator traits (S1 Fig), while urbanization and captivity required more time to reach an asymptote (i.e., changes in antipredator traits were slower in these contexts). Overall, after accounting for the fixed and random terms in the Markov chain Monte Carlo generalized linear mixed model (MCMCglmm) model, antipredator traits were estimated to reach their lowest values after only ca. 30 generations for domesticated animals and after ca. 90 generations for urbanized animals (Fig 1 and S1 Fig). The best model explaining the change in the coefficient of variation (CV) of antipredator traits varied by context. A locally estimated scatterplot smoothing (loess) function without any a priori assumptions fit a Weibull-like curve along which there was an increase in the variance in the first years of exposure to humans (except for captive species), followed by a gradual decrease (Fig 2A). However, the fit was only significant for domesticated species once both fixed and random effects were considered in the MCMCglmm (Fig 2B and S1 Table). Interestingly, the CV did not vary among contexts in generation "0," indicating that the variance in antipredator traits for all wild animals studied were similar. Adding 3 different life-history traits for each species—maximum longevity, foraging guild, and sociality—as fixed effects did not explain the change in CV over time. However, for both domesticated and urbanized species, foraging guild (carnivore, omnivore, herbivore, granivore, or insectivore)

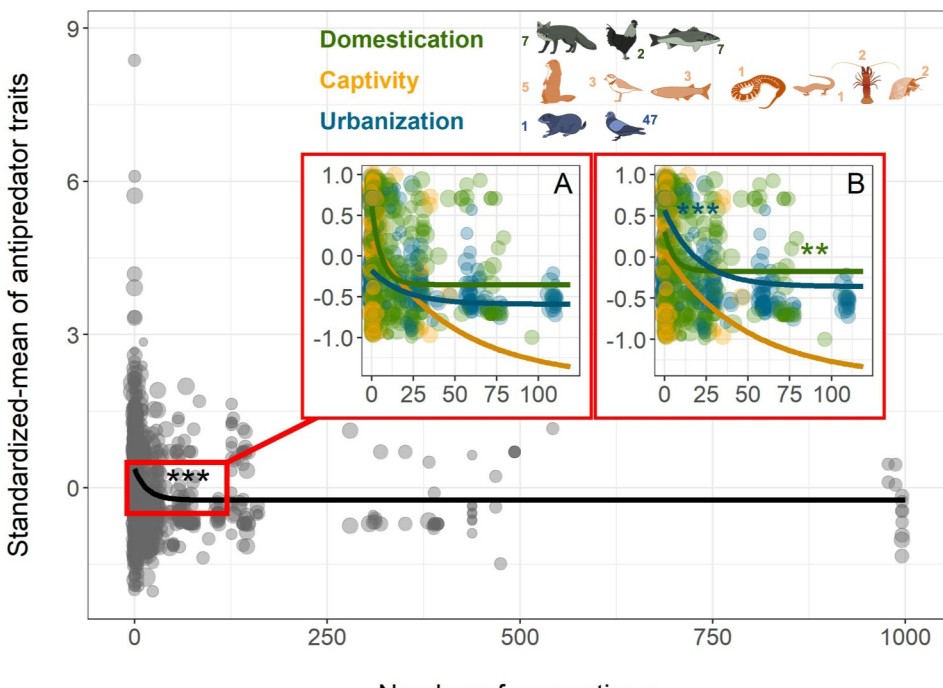

**Fig 1. In multiple species, interactions with humans reduce the overall mean of antipredator traits over time, black line (all contexts).** (A) Lines correspond to the best inverse model fitting the data using the cma-es function. (B) Lines correspond to the best inverse model with the outputs (slopes and intercepts) of the MCMCglmm model for the mean. Dot size is proportional to the log-transformed number of replicates used in each study. "***" indicates significant $p$-value $< 0.001$ and "**" significant $p$-value $< 0.01$ of the inverse slopes with the MCMCglmm model (S1 Table). The number associated with each animal indicates the number of species in each of the following taxa: Mammals, Birds, Reptiles, Fish, Amphibians, Arthropods, and Mollusks. All data and R code supporting the figure are available in S1C Data. Species design: Pierre Lopez (MARBEC). cma-es, covariance matrix adapting evolutionary strategy; MARBEC, Marine Biodiversity, Exploitation and Conservation; MCMCglmm, Markov chain Monte Carlo generalized linear mixed model.

and sociality (gregarious or solitary) significantly improved the models of the mean trait change when added as a social level × generation interaction (Fig 3 and Fig 4; S2 Table and S3 Table). Overall, the inverse slope decreased more rapidly in the first generations for domesticated herbivores compared to both domesticated omnivores and carnivores (Fig 3A and S3 Table, although this result should be interpreted with caution since most of the pattern is driven by mammals (S2 Fig). Similarly, the slope was steepest for urbanized herbivores compared to both insectivores and granivores (Fig 4A and S3 Table). Antipredator responses were lost more quickly under domestication in solitary compared with relatively more gregarious species (Fig 3B and S3 Table), while we detected the opposite pattern for urbanized species (Fig 4B and S3 Table). We note that most of the urbanized species for which we had data were birds, and the majority of these were insectivores or granivores.

## Discussion

For all contexts, a reduction in mean antipredator trait values was observed from the first generation onwards, with the most rapid changes occurring during initial exposure to humans. Such rapid phenotypic modification could be easily explained by genetic evolution in domestic species because of the tendency of humans to select for docile animals and the significant heritability previously observed for these traits in several species [32–34]. Here, we show that

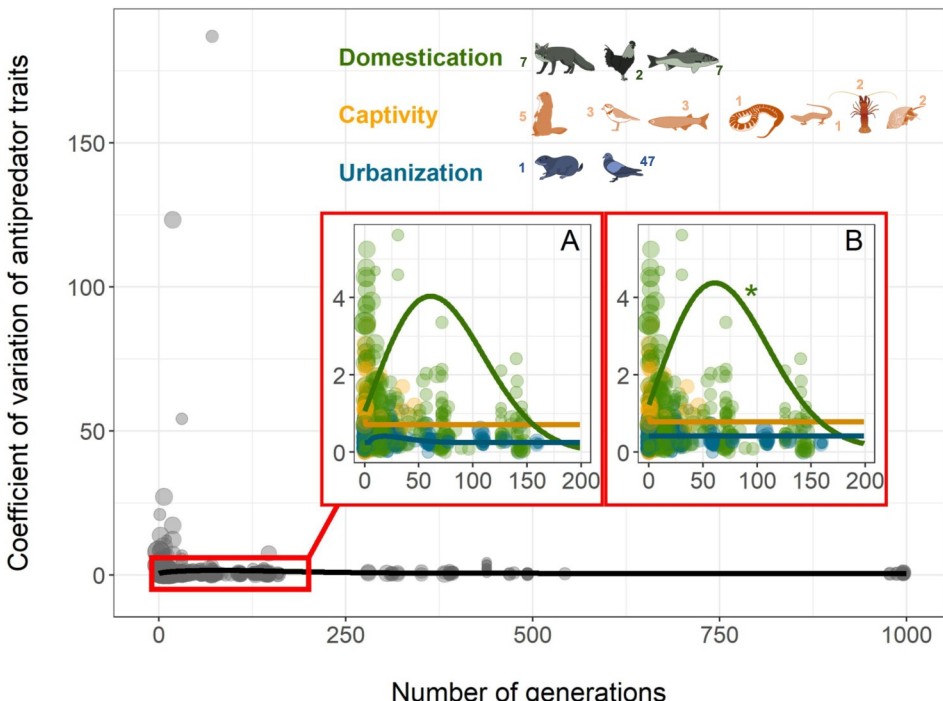

**Fig 2. Effects of human presence over time on the CV in antipredator traits, black line (all contexts).** (A) Lines correspond to the best loess fitting the data using the cma-es function. (B) Lines correspond to the best Weibull model with the outputs (slopes and intercepts) of the MCMCglmm for the CV. Dot size is proportional to the log-transformed number of replicates used in each study. "*" indicates significant $p$-value $< 0.05$ of the Weibull slopes with the MCMCglmm model (S1 Table). The number associated with each animal indicates the number of species in each of the following taxa: Mammals, Birds, Reptiles, Fish, Amphibians, Arthropods, and Mollusks. All data and R code supporting the figure are available in S1C Data. Species design: Pierre Lopez (MARBEC). cma-es, covariance matrix adapting evolutionary strategy; CV, coefficient of variation; MARBEC, Marine Biodiversity, Exploitation and Conservation; MCMCglmm, Markov chain Monte Carlo generalized linear mixed model.

urbanized animals also lose their abilities to respond to predators in a way that is similar to animals that go through the process of domestication, albeit at a slower rate. The same trend was also found for animals in captivity, although this was not statistically significant ($p$ = 0.13), suggesting that a larger sample size is required to better understand the precise dynamics of the loss of antipredator behavior when animals are merely brought into captivity without being actively domesticated.

The phenotypic shift in antipredator traits in response to urbanization is well documented in multiple taxa [35], and 4 non-mutually exclusive mechanisms may underlie this pattern. First, urbanization may create unintentional selection and lead to an evolutionary response [36]. Second, individuals may vary along a continuum of tolerance to humans [37,38], and the increased human presence in urban areas may therefore lead to differential settlement of the more tolerant individuals within the urban environment [21]. Third, behavioral flexibility driven by habituation to humans and human disturbances may increase the tolerance of animals to humans, and this tolerance may be generalized to include a suite of antipredator traits [39–41]. Fourth, the change in antipredator traits may be facilitated by developmental plasticity in which environmental cues animals receive as they develop determines the expression of their traits as adults. The rapid decrease in the means of antipredator traits for our urbanized species suggest that phenotypic plasticity (i.e., either behavioral flexibility or developmental plasticity) is at least partly responsible for observed changes. This supports the findings

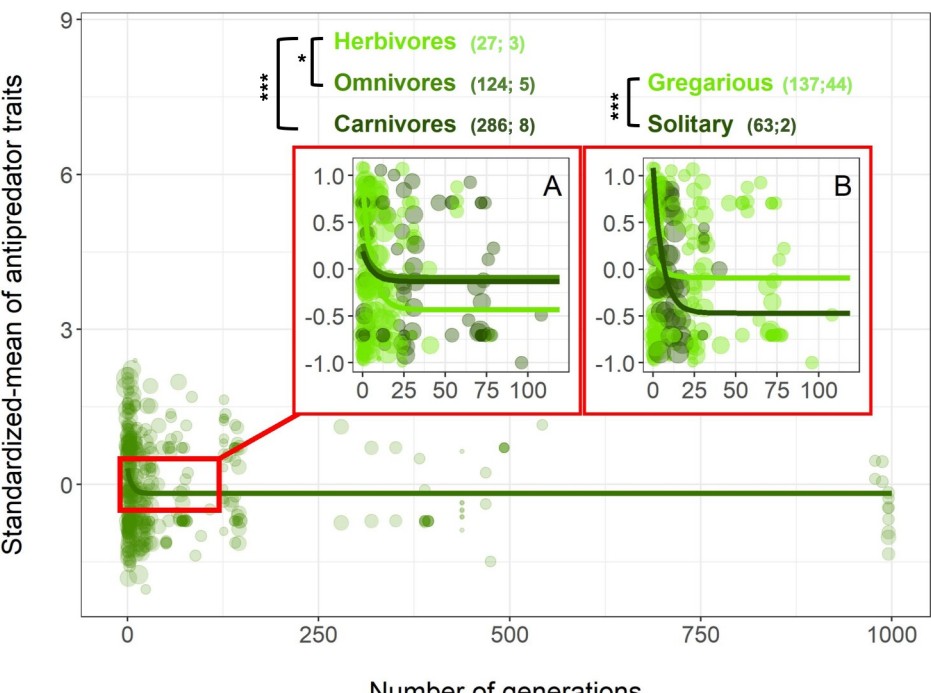

**Fig 3. The decrease in average antipredator traits over generations of domestication differs as a function of animal's foraging type (A) and social status (B).** The lines represent the best inverse model with the outputs (slopes and intercepts) of the MCMCglmm. In (A), model was fitted with foraging type × generation interaction. In (B), model was fitted with social status × generation interaction. Dot size is proportional to the log-transformed number of replicates used in each study. The number in brackets represents the number of estimates followed by the number of species. "*" indicates significant $p$-value < 0.05 and "***" significant $p$-value < 0.001 between slopes within each life-history trait using the MCMCglmm model (S3 Table). All data and R code supporting the figure are available in S1C Data. MCMCglmm, Markov chain Monte Carlo generalized linear mixed model.

reported by Hendry and colleagues [42] showing that phenotypic change in wild animals is greater in anthropogenic contexts compared to natural ones and that this change is mostly driven by phenotypic plasticity.

Both domestication and captivity differ from urbanization by creating an immediate and nearly complete reduction in predation risk. Thus, we expected that domestication and captivity would be the contexts with the strongest effects on the variance in antipredator traits. However, we only found significant changes in the CV of domesticated animals (Fig 2B), suggesting that there might be a key difference between focused selection for domestication the incidental selection we see in captivity and the potentially weaker selection we see in urbanized environments.

The reduction in predation associated with domestication releases constraints on the trade-off between investment of energy in predator defense mechanisms and other life-history traits. As a consequence, a greater variety of phenotypes can be expressed without direct consequences for survivorship, and this relaxation in predation pressure can explain the increased variance over the first generations under domestication. After about 50 generations of selection for domestication, we also observed a general trend for reduced variation in the average antipredator responses. This suggests a shift to another type of selection pressure, which may reflect animals becoming adapted to their new environment along with the directional selection caused by domestication (for example, by humans). It is noteworthy that while the overall antipredator response of domesticated populations stops changing after about 30 generations,

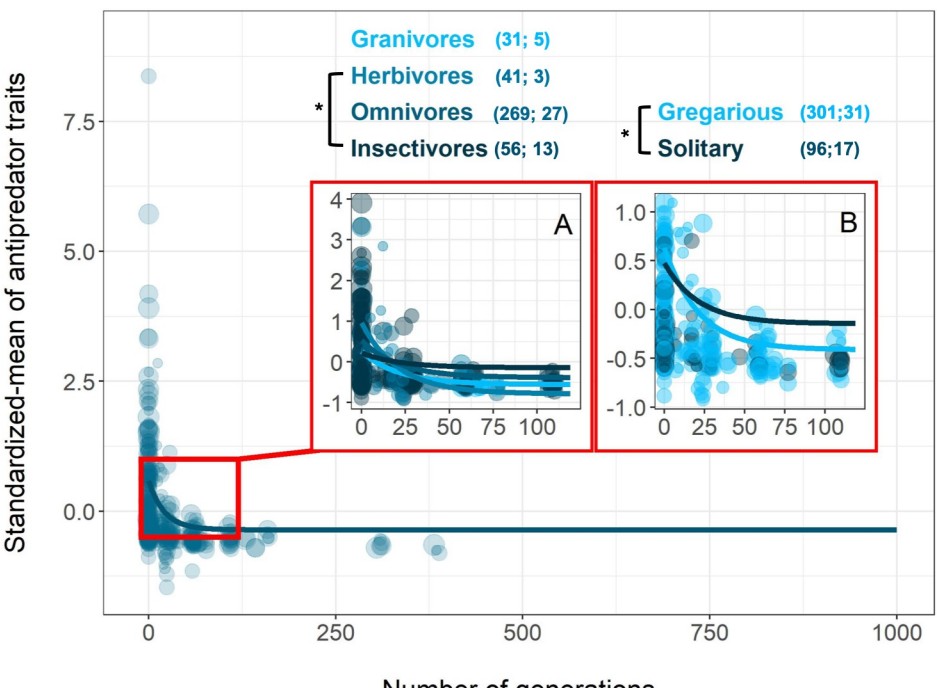

**Fig 4. The decrease in average antipredator traits over generations of urbanization differs according to an animal's foraging type (A) and its social status (B).** The lines represent to the best inverse model with the outputs (slopes and intercepts) of the MCMCglmm. In (A), model was fitted with foraging type × generation interaction. In (B), model was fitted with social status × generation interaction. Dot size is proportional to the log-transformed number of replicates used in each study. The number in brackets represents the number of estimates followed by the number of species. "∗" indicates significant p-value < 0.05 between slopes within each life-history trait using the MCMCglmm model (S3 Table). All data and R code supporting the figure are available in S1C Data. MCMCglmm, Markov chain Monte Carlo generalized linear mixed model.

these populations are still composed of both unwary and wary individuals. Indeed, at that specific stage, the CV of antipredator response is still relatively high. This variation underscores the fact that both individuals with "high fear" and "low fear" (or alternatively, individuals that can exhibit both phenotypes) have been able to survive, likely because of relaxed selection. Those particular phenotypes would then progressively disappear over time, with directional selection favoring docile and robust individuals.

Although not significant for urbanization, a general Weibull-like curve was observed over time (Fig 2A), suggesting similar but weaker effects to the processes occurring in domestication and captivity. The weaker effect seen in urban areas could reflect the great heterogeneity in urban selection pressures (for example, city size, population density, age since establishment, and variation in how urbanization was defined [43]) and the relatively homogenous species composition (birds represented the majority of our estimates). By contrast, our studies of captive animals included a diversity of species studied in diverse situations, which might account for the difficulty of separating the effect of captivity. Future work must isolate specific attributes of both urban and captive environments and quantify their impact on changes in antipredator traits following exposure to human-induced relaxed predation pressure.

Interestingly, the rapid disappearance of fear-related traits for both domesticated and urbanized species was associated with 2 main life-history traits, foraging guild and sociality, but not a third, longevity. Herbivores lost antipredator traits more quickly than omnivores or carnivores for domesticated species. Importantly, this was seen after controlling for variation

explained by potentially shorter generation times. This could reflect the strong selection that predators exerted on herbivores, which, if relaxed by humans, leads to a rapid loss of antipredator responses [44]. Behaving excessively cautiously when there are no longer any predators around might be especially costly for herbivores who allocate considerable time foraging on relatively low-quality food resources [45]. Similarly, solitary species, like solitary individuals, cannot rely on conspecifics for protection and may need to allocate substantial time to antipredator behavior, as exemplified in mammals [46], birds [47], and fishes [48]. For solitary species, the loss of predators may lead to a more immediate loss than in social species, for which vigilance may have social functions as well as antipredator functions and thus be retained under relaxed selection. For urbanized birds, we detected the opposite pattern, in which the decrease in the expression of antipredator traits appeared stronger in gregarious species when compared with solitary species. This may reflect a relatively small sample size or that solitary species can coexist with humans. Future work to identify the mechanism of this unexpected finding is warranted.

Animals are brought into captivity for a variety of reasons, including display in zoos and aquaria and for conservation purposes such as protection from human-induced disturbances such as exotic predators [49] and/or for captive breeding for reintroduction. Once in captivity, antipredator behavior can be lost quickly [50] and may be driven by behavioral flexibility as well as by adaptive and nonadaptive evolutionary changes. The nature of such losses suggest that conservation scientists should pay particular attention to the effects that a predator-free captive environment can have on antipredator traits, especially if future reintroductions are planned [51]. Because these changes can happen quickly and are likely to influence many traits, animals may have to be retaught to display antipredator behaviors [52] or be exposed to predators [53,54] prior to release to prevent the extreme predator-induced mortality often seen following the release of captive-reared animals [55]. Such interventions may lead to both phenotypic modifications and evolutionary changes in a population. Animals surrounding cities or living close to villages and human habitation may become less wary towards humans but also less wary towards genuine predators [56]. Our data-driven conclusions go even further, showing that being in urbanized areas leads to a rate of phenotypic change that is apparently faster than that seen in captive animals that are intentionally isolated from predators. This loss of antipredator traits might have consequences for animal survival in an increasingly human-dominated world.

## Conclusion

In sum, human presence modifies animal antipredator traits in 3 key contexts: urbanization, captivity, and domestication. The patterns were similar; there was an expected overall decrease in mean antipredator traits that was also associated with an increase in variation in these traits over the first generations, followed by a reduction in variation over time. In addition to the ethical issues raised by this human-driven alteration in the evolutionary trajectories of numerous species [57], these changes also make animals particularly vulnerable to actual predators. In the context of HIREC [5], the overall diminished response to predators could have multiple fitness consequences in urbanizing areas [27]. Conserving the variety of antipredator responses that exist within a population will ultimately help sustain it, and this might involve intentionally exposing animals to predators or to predator-related cues for conservation purposes [53,54] to prevent the loss of necessary antipredator traits.

## Materials and methods

The aim of this meta-analysis was to evaluate the mean and the variance in behavioral and physiological antipredator traits as a function of the number of generations under which

animals have lived in urbanized environments (estimated as time since colonization divided by the average time to maturity), the number of generations in captivity, or the number of generations since animals have been domesticated under strong, directional selection.

## Literature search

We conducted 3 separate searches using all databases in Web of Science (Clarivate Analytics, PA, USA). In each of the searches, we combined the same cluster of keywords representing antipredator traits with one keyword representing the type of anthropogenic effect that we were investigating: urbanization, captivity, and domestication (keywords were urban*, captiv*, and domestic*, respectively). The antipredator keywords cluster was ("anti-predator*" OR antipredator* OR fid OR vigilan* OR apprehens* OR mobbing OR "alarm call*" OR "alarm vocal*" OR flee OR fleeing OR tame* OR bold*). The asterisks in the search terms serve as placeholder for any letter or word. The searches were restricted to the following research areas: "environmental sciences ecology," "zoology," "behavioral sciences," "biodiversity conservation," "marine fresh water biology," "science technology other topics," and "evolutionary biology." All searches were conducted on 5 February 2017, using Ben-Gurion University's institutional subscription. Selected studies were categorized in the 3 contexts based on the following definitions:

- Urbanized species are "those that thrive in urban habitats"[58,59].

- Captive species "live under conditions directly imposed by humans through rearing conditions"[60] but are not used for production nor selected for specific traits.

- Domesticated species live under rearing conditions in order to produce and/or are selected.

Hence, based on these definitions and the classification previously made [61,62], we distinguished captive from domesticated species by considering that any species selected for a given purpose (food, fur, tameness, pets, experimentally selected lines) was domesticated. Those only reared with humans (without intention to produce or select for specific traits) were considered captive. All collected data are available in S1A Data.

## Data management and PRISMA diagram

The literature search led to a total of 851, 816, and 1,204 studies for the 3 anthropogenic contexts urbanization, captivity, and domestication, respectively (S1 PRISMA Checklist). After removing all irrelevant studies (i.e., those that did not evaluate antipredator traits, those that were not journal articles), the data set included 7, 59, and 107 studies, respectively. Within each context, we only retained studies for which there were at least 2 values per trait (i.e., >1 generation). Dogs were removed from the domestication data set because of the very high variability observed (between breeds) due to repeated bouts of selective breeding for different traits (S1 PRISMA Checklist). For all studies, we extracted the average value and the standard deviation of the antipredator trait and contacted authors when data were unavailable. For the means, 90 values were removed because they originated from studies investigating antipredator traits for animals selected for increasing antipredator phenotypes (high fear or aggressiveness). Altogether, this led to a total of 404, 208, and 437 means and 404, 174, and 527 standard deviations of interest for studies related to urbanization, captivity, and domestication, respectively (S1 PRISMA Checklist). Overall, each study had an average of 4.7 mean values (SD ±4.5) and investigated 1.6 antipredator traits (SD ±1.1). A table summarizing all traits investigated per context is presented in S4 Table. Most studies were from Europe and North America (S3 Fig). We used the "worldHires" function from the "mapdata" R package, which

provides a legacy world database containing approximately 2 million points representing the world coastlines and national boundaries. We then create the base map with the "maps" package. We examined mammals, birds, reptiles, fish, and mollusks. These steps are illustrated in the PRISMA diagram (S1 PRISMA Checklist). All collected data and R code are available in S1B Data.

## Data transformation

The CV was calculated by dividing the standard deviation by the mean for all antipredator traits (which corresponds to the spread of data at the between-individual level). Since the mean was sometimes negative, we used the absolute value of the CV in the analysis. The means were first transformed so that larger values always reflected greater expression of antipredator traits. This required using the inverse of values when the measured trait was inversely proportional to the greatest expression of an antipredator trait. For example, high values of "latency before escaping a predator" translated to reduced antipredator capacities, and thus required it to be inversed (by multiplying by −1). Means were then all shifted so that they were >0 by adding the absolute minimum value from the trait so as to have a baseline value of 0 for all traits. Then, we rescaled all means to a common scale by subtracting each observed mean value from the trait's mean within contexts and studies and then dividing this value by the associated standard deviation. The final mean value used is referred to the "standardized mean" in the text. All collected data and R code are available in S1B Data.

## Life-history traits

For each species available in the final data set, we collected from multiple online databases (for example, Animal Diversity Website) information regarding the foraging guild, maximum longevity, and the social level. These collected data and the associated references are provided in S5 Table.

## Phylogenies

We created the phylogeny using the *rotl* package [63], which provides an interface to the "Open Tree of Life" [64], in which most of our species were available. For the few species (urbanization: $n = 4$; captivity: $n = 1$) in our data set that were not included in this phylogeny, we used a closely related (congeneric) species as a substitute [65]. The trees were pruned using the R package *picante* 1.6–2 [66]. Branch length was obtained from the "Open Tree of Life" [64] providing ultrametric trees. The phylogenetic trees of the taxa included in the study are provided in S4A–S4C Fig.

## Explaining the pattern of variation in the mean

We first fitted a locally estimated scatterplot smoothing (loess) function through the means across generations for all contexts together and separately. This approach allowed us to describe the overall relationship that best fit the data, without any a priori assumptions. For the entire data set and for each context, the loess (with the default parameters) described an overall strong decrease in fear-related traits in the first generations. We thus fitted a curve similar to an inverse model for each context with the equation

$$f(x) = y_{ini} + k \times a^x,$$

where "$y_{ini}$" is the asymptotic value, "$k$" is the scaling coefficient, and "$a$" a decay factor. A covariance matrix adapting evolutionary strategy (cma-es) was used to optimize parameters of

the inverse function, using the cma-es package [69]. We thus obtained 3 parameters—"$y_{ini}$," "$k$," and "$a$"—that best fitted all points. We then tested the significance of the fit (that fortuitously coincided with our biological predictions) by including those parameters in a MCMCglmm model. If the posterior mean (intercept) was different from 0, then the inverse curve underlining our biological assumption was considered to significantly fit our predictions. For models fitted to the mean and the CV, we set the number of iterations to 1,000,000, using an initial burn in value of 30,000 and a thinning interval of 100. The observations were weighted by the inverse of the sample size. We defined priors necessary for the Bayesian modelling with inverse-Wishart distribution for the variance structure using parameter settings for noninformative priors (expected variance, $V = 1$; degree of belief, $\nu = 0.02$). The trace and distribution of all variables were checked visually, as well as the autocorrelation between iterations. Each model was run at least 5 times to check for the consistency of the results (including parameter estimates and Deviance Information Criterion [DIC]). All data and R code are available in S1C Data.

## Explaining the pattern of variation in the CV

A similar approach was used to model the absolute CV across generations. We searched the best smoothing parameter of the loess using a loop and the cma-es function. This allowed us to obtain loess splines that fitted our initial biological predictions (strong increase and then a decrease of the variance over the generations), so that a Weibull model with the following equation has been considered:

$$f(x) = y_{ini} + a\frac{k}{\lambda}\left(\frac{x - t_{min}}{\lambda}\right)^{(k-1)} e^{-\left(\frac{x - t_{min}}{\lambda}\right)^k},$$

where $y_{ini}$ is the baseline value, $a$ and $k$ are scaling coefficients, $\lambda$ is the shape coefficient, and $t_{min}$ is the start of the change.

We ran the cma-es function until we identified that with the lowest mean absolute error (MAE). Those best 5 Weibull parameters were then included in MCMCglmm to obtain posterior distribution values for the coefficients (intercept and slopes of the Weibull) and test for their significance (if the 95% credible intervals [CIs] of the estimates of intercept and slope do not span zero, they are considered statistically significant). We then compared models with different random effects and based on their DIC using the MuMIn package [68], from a full model with "species," "phylogeny," "studies," and "traits type" to a model containing only "traits type" as a random effect. The trait type (physiological or behavioral) appeared to have a major effect for both mean and CV (S6 Table). We first compared all models with different random factors. Since the outputs of the models were similar, we only kept the full model (4 random factors) for further analysis with life-history traits so that we could describe global pattern independently of the phylogenetic relationship between species.

## Testing the effects of life-history traits

The 3 investigated traits (i.e., foraging guild, social level, maximum longevity) and their associated interactions with generations were tested in a complete model fitted to the best smoothing found for the means and CVs (see above) for each of the 3 contexts: domestication, urbanization, and captivity. For domesticated and urbanized species (for which the simple inverse model was significant), we removed 1 fixed effect at a time and classified all models based on their DIC using the MuMIn package (S2 Table). To illustrate these effects, we refitted the model with only a single life-history trait and used the outputs of the MCMCglmm models to plot the effect.

## Controlling for heterogeneity

We used MCMCglmm to partition the total heterogeneity $I^2$ among different sources: variation explained by study identity, species identity, and species relatedness (phylogeny) and by residual variation (i.e., remaining to be explained by the predictor variables) [69] (S6 Table). We calculated the degree of phylogenetic signal in our effect size estimates using the phylogenetic heritability index, $H^2$, which is the variance attributable to phylogeny in relation to the total variance linked to species expected in the data [69]. $H^2$ is equivalent to Pagel's lambda coefficient [70], where values close to 1 are associated with strong phylogenetic signals as opposed to values close to 0. All data and R code are available in S1C Data.

## Supporting information

**S1 PRISMA Checklist. PRISMA flow diagram describing literature search, selection, and analyses.** The orange rectangles narrow the numbers of studies per context prior to the analysis steps (purple rectangles) performed on the multiple means and CVs of antipredator traits within each study. Numbers are provided per context: red for urbanization; blue for captivity; and green for domestication. CV, coefficient of variation; PRISMA, Preferred Reporting Items for Systematic Reviews and Meta-Analyses
(TIF)

**S1 Fig. Derivatives of changes observed over generations for the mean of antipredator traits in the 3 contexts of human presence.** Values are expressed as a rate of change per generation (/g). Lines correspond to the derivatives for the best inverse model fitting the data using the cma-es function. All data and R code supporting the figure are available in S1C Data. cma-es, covariance matrix adapting evolutionary strategy.
(TIF)

**S2 Fig. The decrease in average antipredator traits over generations of domestication differs as a function of animal's taxon.** The lines represent the best inverse model with the outputs (slopes and intercepts) of the MCMCglmm. Dot size is proportional to the log-transformed number of replicates used in each study. All data and R code supporting the figure are available in S1C Data. MCMCglmm, Markov chain Monte Carlo generalized linear mixed model.
(TIF)

**S3 Fig. Origin of the data used for the meta-analysis.** Circle size is proportional to the number of studies. Some studies were simultaneously performed at several different sites. The map was designed with the open source software R (3.3.0): package "maps" and "mapdata." All data and R code supporting the figure are available in S1C Data.
(TIF)

**S4 Fig.  Phylogenetic trees of animals used in the phylogenetic meta-analysis for (A) domesticated, (B) captive, and (C) urbanized species.** All data and R code supporting the figure are available in S1C Data.
(TIFF)

**S1 Table. Meta-analytic estimates of the slopes (fitted using MCMCglmm) of factors that best describe how generation time affects normalized-mean and CV of antipredator traits using 4 random effects: "species," "phylogeny," "studies," and "trait type".** Inverse and Weibull models are presented for all contexts together and also analyzed separately: domestication, captivity, and urbanization. Inverse models are presented for all contexts together and also analyzed separately: domestication, captivity, and urbanization. Parameters for the model

were obtained using the covariance matrix adapting evolutionary strategy (cma-es package) [67]. Estimates with 95% CIs not spanning zero are considered statistically significant (significance codes: <0 "***," 0.001< "**," 0.01< "*," 0.05< ".," 0.1< blank). Only the intercept of the "urbanization" model was significant (posterior mean value = 0.91, *p*-value = 0.004). CI, credible interval; cma-es, covariance matrix adapting evolutionary strategy; CV, coefficient of variation; MCMCglmm, Markov chain Monte Carlo generalized linear mixed model.
(XLSX)

**S2 Table. Comparison of the models presenting all combinations of life-history traits "foraging guild," "social level," and "longevity maximum" for urbanized and domesticated contexts.** These models were compared based on their DIC using the MuMIn package [68], which also provides the weight of each model. DIC, Deviance Information Criterion.
(XLSX)

**S3 Table. Meta-analytic estimates of the intercept and slopes (fitted using MCMCglmm) of factors that best describe how generation time affects normalized-mean of antipredator traits for the best models (model1d and model1u) described in S2 Table.** For both domesticated and urbanized contexts, the interaction between generation (inverse parameters) and 2 life-history traits, "foraging guild" and "social level," are described. The number (*n*) of estimates and species for each category is also provided. MCMCglmm, Markov chain Monte Carlo generalized linear mixed model.
(XLSX)

**S4 Table. The different antipredator traits investigated in our study.**
(CSV)

**S5 Table. Life-history traits of the species used in the analyses.**
(XLSX)

**S6 Table. Estimates of heterogeneity $I^2$ (%) from the normalized-mean meta-analysis and the CV meta-analysis for the best model with all random factors for the 3 different contexts and all contexts.** The MLPMA allowed partitioning $I^2$ among varying levels. Posterior mean and upper and lower 95% CIs for each estimate of $I^2$ are provided. $H^2$ is phylogenetic heritability, the proportion of variability in the data attributable to the phylogenetic component of MLPMA calculated following Nakagawa and Santos [71]. CI, credible interval; CV, coefficient of variation; MLPMA, multilevel phylogenetic meta-analysis.
(XLSX)

**S1 Data.** All data for the meta-analysis, (A) initial data management, (B) data management, and (C) R code and data.
(ZIP)

## Acknowledgments

We thank Diogo S. M. Samia for previous fruitful discussions on the subject. We also thank Pierre Lopez for kindly drawing all species.

## Author Contributions

**Conceptualization:** Benjamin Geffroy, Bastien Sadoul, Daniel T. Blumstein.

**Data curation:** Benjamin Geffroy, Bastien Sadoul, Breanna J. Putman, Oded Berger-Tal, Anders Pape Møller.

**Formal analysis:** Benjamin Geffroy, Bastien Sadoul, László Zsolt Garamszegi, Anders Pape Møller.

**Methodology:** Benjamin Geffroy, Bastien Sadoul, László Zsolt Garamszegi, Anders Pape Møller.

**Project administration:** Benjamin Geffroy.

**Writing – original draft:** Benjamin Geffroy, Bastien Sadoul, Daniel T. Blumstein.

**Writing – review & editing:** Breanna J. Putman, Oded Berger-Tal, László Zsolt Garamszegi, Anders Pape Møller.

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
