## [Editor Report · Decision Letter 0]

2 Mar 2020

Dear Dr Geffroy, 

Thank you for submitting your manuscript entitled "Unravelling the evolutionary dynamics of antipredator traits following exposure to humans" for consideration as a Research Article by PLOS Biology.

Your manuscript has now been evaluated by the PLOS Biology editorial staff, as well as by an academic editor with relevant expertise, and I'm writing to let you know that we would like to send your submission out for external peer review.

Please re-submit your manuscript within two working days, i.e. by Mar 04 2020 11:59PM.

Kind regards,

Roli Roberts

Senior Editor

PLOS Biology

---

## [Decision Letter · Decision Letter 1]

16 Apr 2020

Dear Dr Geffroy,

Thank you very much for submitting your manuscript "Unravelling the evolutionary dynamics of antipredator traits following exposure to humans" for consideration as a Research Article at PLOS Biology. Your manuscript has been evaluated by the PLOS Biology editors, an Academic Editor with relevant expertise, and by three independent reviewers. Thank you for your patience while we processed your manuscript in the current difficult circumstances.

You'll see that the reviewers are broadly positive about your study, but raise a number of concerns that will need to be addressed. These include a need to describe the database and analyses more clearly, and also to explain the objectives & implications of this analysis better. There are also suggestions of further analyses that might be appropriate and which might strengthen the study further.

In light of the reviews (below), we will not be able to accept the current version of the manuscript, but we would welcome re-submission of a much-revised version that takes into account the reviewers' comments. We cannot make any decision about publication until we have seen the revised manuscript and your response to the reviewers' comments. Your revised manuscript is also likely to be sent for further evaluation by the reviewers.

We expect to receive your revised manuscript within 2 months. 

**IMPORTANT - SUBMITTING YOUR REVISION**

*Re-submission Checklist*

*Published Peer Review*

*PLOS Data Policy*

*Blot and Gel Data Policy*

Sincerely,

Roli Roberts

Senior Editor

PLOS Biology

REVIEWERS' COMMENTS:

Reviewer #1:

This MS addresses an important topic in an era of human transformation of the planet. Many species have traits that can be subject to new evolutionary ecological pressure. Developing an understanding of the consequences of those new evolutionary ecological pressures on species is necessary to understand the fate of biodiversity. 

The MS reports on a meta-analysis of the loss of anti-predator responses of prey species subject to different kinds of human agency and pressure (i.e., urbanization, captivity, and domestication). It compares rates of loss of anti-predator responses across numerous behavioral and physiological traits among the three different human agencies/pressures. It concludes that there is rapid loss in trait responses associated with all three human agencies. While the findings are potentially interesting, I find that I cannot judge whether or not the results are scientifically robust. This is because the presentation of methods and data are inadequate to understand the reliability of the data used in the analyses. At the very least, the MS fails to meet my litmus test that the procedures used in the MS are sufficiently explained that I could repeat the study independently without consulting the authors. 

The MS states (beginning line 235) that there were 61, 59, and 107 studies that provide trait response data for urbanization, captivity and domestication respectively. But later (beginning line 243) the narrative states that 417, 206 and 437 means were estimated from those studies. Clearly there is a lack of independence in the data here. But it is unclear what that lack of independence stems from. Is it due to time series of trait changes presented in the different studies; or is it due to multiple traits measured in the same study; or is it a combination of the two? This needs clarification. As well, how the lack of independence is dealt with in the data analyses needs better explanation. Moreover, it is unclear to me how you can estimate 437 means for domestication yet have 527 standard deviations.

It would also help to understand how the studies from which the meta-data were extracted were generally done. That is, were they experiments that challenged prey with risk or risk cues relative to proper controls? Were they field or laboratory-based studies? How did the experiment risk or risk cue compare in magnitude to risk under "wild" conditions. How was the number of generations of human exposure determined for a species used in a particular study?

Analyses of trait response effects use different kinds of traits for urbanization, captivity and domestication (Table S1). Analyses for urbanization seem to be based exclusively on behavioral traits. The other two are based on behavioral and physiological responses combined. How do we know that conclusions about differences in the magnitude of trait responses is due to the form of human agency (as concluded in the MS); as opposed to the use of different kinds of traits, or even an imbalance in the relative number of physiological and behavioral traits used for each form of agency that may confound the comparison? This concern stems from the observation (line 306) that behavioral and physiological trait responses can each have significant effects on mean and variance. 

Beyond identifying broad taxonomic groups, the narrative in the MS is vague on what particular species were used in the data analyses and where the data for each species falls along the generation time continuum (x-axis). My concern is that with any species that has at least a generation time of a year or more, how does one get data for 120 generations (let alone 1000 generations) when it is unlikely that the scientific studies presenting the trait data likely weren't even published that long ago. Perhaps the data presented in Fig. 2 are cross-sectional, using different species and the number of generations that they have been associated with humans. But how do we robustly know that a species that has, say been associated with humans for 1000 generations, behaved differently before human contact? Also, if data for different species fall on different points along the generation time axis, how do we know that data for the heightened trait responses when species have only recently been associated with humans (data near origin in Fig. 2) aren't for species that are naturally more apprehensive to predators than other species that may fall in the tails of the axis? What I am driving at is how are the data for each of the 16 domesticated, 16 captive and 50 urbanized species distributed along the x-axes of Fig. 2? All of this is to say that I have concerns about how the study "controls" for human effects, in the sense of robustly knowing what the species' trait responses were before being exposed or subjected to human effects and what the responses were gradually over the many generations associated with humans. 

The sources of the data are presented in the supplemental file ALL.csv. But the information in that table is not presented transparently nor is there an adequate description of what all of the column headers mean. Hence it is very difficult to verify the sources of the data. 

Additional editorial comments

Line 67—problem with logic. The sentence states that prior to human contact species have been under strong selection to avoid predation. This sentence could be interpreted this way: If humans didn't contact the species, how do we know that they were or weren't under strong selection, because there were no humans around to measure things. Please consider rephrasing. 

Line 71—expand on the conceptual ideas from evolutionary biology for why knowing patterns of variation in anti-predator responses is even important. 

Line 74-75—this explanation of "human shield" is inaccurate. The original conception says nothing about reduction in the number and diversity of predators. The original conception merely states that prey that associate with humans or human infrastructure may be protected from predation because of the deterrent effects of human presence. In the human shields idea, predators merely spatially disassociate from prey that are near humans or human infrastructure. 

Line 75-76—this sweeping claim that predation risk is essentially eliminated for domesticated animals is also inaccurate. Many domesticated cattle and sheep are subject to predation—the cause of much human-carnivore conflict (hence predation pressure is not relaxed)—and those domesticated animals also often exhibit anti-predator behavior. This statement needs some qualification. 

Line 80-85—the narrative says that the study will test the "relaxed-selection hypothesis". But in the discussion, it also invokes phenotypic plasticity as a possible explanation for trait changes. This means that the study is not truly testing for relaxed selection—in fact as described, the data cannot speak to this because it would require conducting reciprocal transplant experiments to test for adaptation. The study can only say that the trait responses to predation challenges seem to diminish over time either from phenotypic plasticity or form altered selection. 

Reviewer #2:

The authors, an all-star group of behavioral ecologists with a history of studying antipredator behavior and/or applying behavioral insights to conservation biology, conducted a meta-analysis looking at changes in the mean and coefficient of variation in antipredator traits (I will call it fear) following exposure to humans in the context of domestication, captivity and urbanization. This is a valuable endeavor. Taking a devil's advocate view, however, one could say that showing that animals have generally become less fearful after generations of exposure to humans is obvious, and even the fact that this has happened particularly rapidly in domesticated animals seems obvious. Along these lines, the authors' suggestion that "conservation scientists should pay attention to effects of a predator-free captive environment on antipredator traits, especially if captivity aims at a future reintroduction" is something that is already an established part of conventional wisdom, and the notion that they might need to be taught or exposed to predators is also well known and widely discussed. Still, although some of their main results or conclusions are obvious, it is reasonably valuable to test them with a meta-analysis.

While it is useful to do a meta-analysis to confirm these obvious trends, to me, a more interesting analysis (i.e., one that could tell us things we didn't already know), might be one that explains the variation in responses over time: why did some captive or domesticated species show almost immediate decreases in fear, while others initially retained their fear and took dozens or even 100 generations to become more fearful. The very rapid decrease in fear - where numerous examples exhibited the maximum observed decrease in antipredator traits (-1) essentially immediately (figure 2A) - presumably primarily represents plasticity. Conversely, even among domesticated species, figure 2 also shows some that are still quite fearful after 60-100 generations! Besides the three main types of exposure to humans, what explains this striking variation in initial or long-term response? An obvious idea is that the initial rapid response reflects plasticity, while species that took dozens or more generations to become less fearful have relied more on evolution. Does their meta-analysis yield insights on why some species showed rapid plasticity (immediate learning, developmental plasticity or transgenerational plasticity perhaps involving social learning) while others do not, but instead required many generations to evolve a loss of fear even when held in conditions with no risk? Why did some taxa not evolve a loss of fear even after so many generations without being attacked? Is this related to something about the organisms: their trophic level (do predators lose fear more rapidly than prey?), taxonomic 'type' (do 'higher' vertebrates learn (individually or socially) to be less fearful more rapidly than herps or invertebrates?), activity type (do more active species learn more rapidly than sit-wait species?), social system (do more social species learn via social facilitation more rapidly than solitary species?). Or does it relate to ecosystem type (e.g., the openness of the habitat - others have noted the difference between dynamics in open water vs vegetated zones, or between terrestrial habitats with less vs more vegetation structure)? Or, is it related to the type of captivity or domestication (breeding for productivity as opposed to for work)? Perhaps the authors do not have enough studies to address these interesting questions, but they are the ones that to me, would make for a more exciting paper! To be a bit more specific, I am picturing analyses of the residuals, the deviations from the fitted functions of fear over time. I realize that these analyses may be non-trivial, but still, if the authors do not add these analyses to this paper, I could imagine asking for access to their data set to do the analyses myself. I am very curious to know what factors might explain which species or situations exhibit more rapid vs slower responses to living with humans! 

In this context, it is worth noting that a classic paper by Hendry et al. 2008 Molecular Ecology 17:20-29 examined effects of humans on phenotypic change and found that many examples exhibited very rapid change that suggests that plasticity is very important. Perhaps I did not look carefully enough, but the authors did not apparently cite this important, relevant paper.

The fact that domestication produced a decrease in fear that reached an asymptote after only about 25 generations, but that the coefficient of variation continued to increase very substantially for another 30 generations or so seems difficult to explain. Logically, I can understand an initial increase in CV even before the mean has decreased much; that is, I can imagine a population of fearful animals adding (under relaxed selection) more and more less fearful individuals with initially little decrease in the number of fearful ones, but why would the CV continue to increase even after the mean has become substantially less fearful? The authors discussed this, but I did not find their discussion illuminating.

Figure 1 is not easy to interpret. The y-axis is the occurrence of the two extremes with the black line representing fearful individuals and white lines the fearless ones. Since the black line starts at only an intermediate level (as opposed to a maximum level), I am guessing that by "occurrence", the authors mean the absolute abundance of the types and not the proportional (relative) frequency. Please clarify the meaning of "occurrence". To me, it seems unfortunate to simplify the world into two types. The animal personality literature has been criticized for implying that there are two behavioral types (here, high fear and low fear) as opposed to the reality of a continuous distribution. Perpetuating this problem seems like a bad idea. With urbanization and captivity, the authors suggest that the fearful animals decrease in abundance, while after a while, fearless ones spontaneously appear and increase in prevalence. But in domestication, the fearful animals do not decrease, but instead also increase rapidly in occurrence. Do the authors really mean to suggest these patterns of change for these two behavioral types? If nothing else, they should explain their logic behind the patterns in figure 1 either in the legend or the text. In all three situations, the fearful ones blend into fearless ones (the color of the line goes from black to grey and the line then joins the white line; perhaps this reflects plasticity?), as opposed to the frequency of fearful ones (the black line) continuing to decrease over time. Is this really what the authors have in mind for the dynamics of trait change? Some explanation would be useful. With urbanization and captivity, overall population size continues to increase more or less exponentially (though faster in captivity) while in domestication, it asymptotes. Do the authors really mean to suggest these contrasting population dynamics? The human shield effect is shown to gradually increase smoothly with urbanization and captivity, though more quickly in captivity, as opposed to an abrupt increase in domestication. In all three cases, the color intensity suggests that after some time, the human shield gets equally strong. In contrast, based on the text, one might think that all three situations change abruptly (not gradually over time) from 'natural' to human-altered, but that the magnitude of the human shield differs among situations. 

The bottom part of figure 1 is labelled as "cumulated variance". I am not sure what this means. Is variance piling up over time - are the authors suggesting that it is meaningful to add up variance across time? What does that mean? Or, perhaps they just mean expected variance - and the general notion (as stated on lines 84-85) that with relaxed selection, the population might harbor greater variance in fearfulness. However, that would seem to suggest greater variance allowed with a stronger human shield, but the graph shows the prediction that with a stronger human shield, the cumulated variance increases faster, but to a lower level. This prediction needs further explanation either in the text or in the figure legend.

Overall, I suggest a complete reworking of figure 1.

A few minor points

Please standardize spelling: urbanization vs urbanization.

Line 70: odd to refer to antipredator behavior as being at a "context-dependent equilibrium". While the behaviors are no doubt context-dependent, they are not usually referred to as being at an equilibrium.

Line 241. The authors removed 90 examples of where animals were selected for increased fear or aggressiveness. It is probably OK to remove these examples from THIS paper, but it would be interesting to contrast the efficacy of responses to selection for reduced vs increased fear.

Line 256. Data were standardized. In that context, why do domesticated animals appear to have a much higher mean level of fear in generation zero than captive or urbanized species? Statistically, the fact that domesticated animals were more fearful when first brought into captivity seems to explain why they exhibited a larger, more rapid decrease in fear. But shouldn't all groups have started out with a mean fear level close to zero? I am clearly not understanding something about what figure 2 shows.

Overall, this is an interesting and valuable paper, but could be made somewhat more clear (in particular, figure 1), and much more valuable with additional analyses.

Reviewer #3:

I think that the strengths of this paper relate to the phylogenetic meta-analysis. That strikes me as a very novel contribution to the broad field of study which is seeking to understand how animal communities are changing in light of human behavior. However, the scope of the study is not well-presented leaving the reader to wonder what is the point. 

The paper, at present is a bit under-referenced. There is so much literature on this broader topic, I was surprised to see that the authors only cited 44 papers and just 14 in the introduction. However, this may be because of formatting requirements in PLOS Biology. If so, then please disregard my comment. However, if there is not a strict reference limit for this paper, then the introduction, in particular, feels very under-developed. Several paragraphs in that section, for instance, are just 3-4 sentences in length. It seems that the authors are taking for granted that the readers automatically 'get' what they are proposing to study. I get it, but would like to be supported in the narrative so that my attention continues to be allocated to that which the authors want me to understand. 

More broadly, it is unclear what new information derives from this manuscript. The presentation of the role of this paper by the authors makes it challenging to interpret the values of the results. We already know that urbanization and domestication reduce the intensity of anti-predator traits. The authors admit this in the discussion section when referencing the broader literature. What would be the consequences of reduced anti-predator traits in captive animal populations? After all, animals in captivity will likely never be released into the wild. Thus, there is no real need for anti-predator traits and the reduction of those traits may actually present advantages to people that manage captive facilities (i.e. increased docile nature). Thus, the recommendations emanating from this paper seem odd. The authors write on line 196 that society might consider "intentionally exposing animals in captivity for conservation purposes to predators or predator-related cues." How would that it any way be a conservation-oriented decision? Furthermore, there would be profound ethical dimensions that would be necessary to unpack before a decision like that could be made. 

Line 62: Undoubtedly, all animals have been effected by human disturbance on planet earth. However, it would seem that the structure of this sentence is most applicable to large vertebrates. I recommend that the authors look specifically at the generality or specifity of their writing in recognition of the fact that sentences like this don't well apply to all animals. 

Line 67: This selective pressure persists in the Anthropocene. I believe that authors are suggesting that the strength of these effects may modulate in line with sustained human interaction. But that principle is not well-conveyed in this sentence. 

Line 70: Do the authors mean "allocation of antipredator"

Line 67: After reading this paragraph, I would recommend that the authors present this information in the absence of human interaction (i.e., the functioning of non-human predator-prey systems). Then the transition to the modulation of these interaction effects via humans can be presented in the following paragraph. 

Line 74: Some setup identifying that the interest here is in three human things (as done so in the abstract) including urbanisation, captivity, and domestication is needed. Furthermore, these are immensely complex relationships that will be stronger for some animal species over others. The reader needs to understand which species the authors are planning to focus on. The authors also switch between urbanisation and urbanization. There is a need to standardize. 

Line 84: This is the first suggestion of fear-related traits. The reader has been prepared to think that the focus is on anti-predator traits. Via that logic, the reader is thinking that the focus of this paper might be on prey species and perhaps large ones that are more likely to be affected by humans. However, fear-related traits could also apply to predator species. Thus, I am not quite confused. The authors need to help the reader understand what they are focusing on and describe how anti-predator and fear-based traits vary. 

Line 95: The authors need to remind the reader which human influences (urbanisation, captivity, and domestication) they are referring to. Also, here again, we have indication of confusing variation in the terminology used by the authors as they once again refer to antipredator. Now we have antipredator traits, antipredator responses, antipredator phenotypes, fear-related traits, high fear, low fear, and fearlessness. The authors really need to standardize their presentation so that the reader can understand what they are trying to do. 

Figure 2: When the authors say 'interactions with humans' do they refer to urbanisation, captivity, and domestication? Interactions with humans is extremely broad and the magnitude of effects would be predicted to be highly variable. 

Line 129: Which 'contexts'? The authors need to do a much better job of developing a narrative flow to help the reader interpret this manuscript. 

Line 146: I fail to see how domestication would result in a 'complete reduction in predation risk.' Many domestic animals still live in systems that are under the risk of depredation. Take for instance, the vast literature associated with human-carnivore conflict built upon carnivore depredation of livestock. Thus, the inference made by the authors on the next sentence is simply inaccurate. 

Line 172: Aquaria? Do the results apply to aquatic species? How so? Again, it is very difficult to interpret which species this analysis is applicable to. 

Line 183: It is the authors responsibility to describe the implications of their results. They mention 'drastic consequences' that might emanate from the knowledge within this study. What are those?

Line 196: Why would society be inclined to retain anti-predator behaviors in captive animals via exposing them to fear? What would be the ethical implications of such a design? I fail to see the logic here.

---

## [Decision Letter · Decision Letter 2]

22 Jun 2020

Dear Dr Geffroy,

Thank you for submitting your revised Research Article entitled "Unravelling the evolutionary dynamics of antipredator traits following exposure to humans" for publication in PLOS Biology. I have now obtained advice from two of the original reviewers and have discussed their comments with the Academic Editor. 

Based on the reviews, we will probably accept this manuscript for publication, assuming that you will modify the manuscript to address the remaining points raised by reviewer #2 and the Academic Editor. Please also make sure to address the data and other policy-related requests noted at the end of this email.

IMPORTANT:

a) Reviewer #2 suggests a number of ways in which s/he thinks you could enhance the appeal of your paper. They ultimately leave this optional, but we and the Academic Editor would like to emphasise the need to do this, especially for the broad readership of our journal.

b) In addition, the Academic Editor has made some suggestions for improving the Title and Abstract (see foot of this email); please implement these.

c) Please attend to my Data Policy requests further down. If may be that your zipped folder already contains the data directly underlying the main and supplementary Figures, but this needs to be made clearer by citing the relevant files in the individual Figure legends.

We expect to receive your revised manuscript within two weeks. Your revisions should address the specific points made by each reviewer. In addition to the remaining revisions and before we will be able to formally accept your manuscript and consider it "in press", we also need to ensure that your article conforms to our guidelines. A member of our team will be in touch shortly with a set of requests. As we can't proceed until these requirements are met, your swift response will help prevent delays to publication.

*Copyediting*

*Published Peer Review History*

*Early Version*

*Submitting Your Revision*

Sincerely,

Roli Roberts

Senior Editor

PLOS Biology

DATA POLICY:

Many thanks for providing your raw data and code in the zipped supplementary folder. In addition, we ask that all individual quantitative observations that underlie the data summarized in the figures and results of your paper be made available in one of the following forms:

Regardless of the method selected, please ensure that you provide the individual numerical values that underlie the summary data displayed in the following figure panels as they are essential for readers to assess your analysis and to reproduce it: the data for the scatterplots in Figs 1, 2, 3, 4, S2, the graph in S1, and the values projected onto the trees in Figs S5ABC. It may be that these are all provided as part of the deposition in your zipped folder, but if so, this needs to be made clearer to us and to the readers. NOTE: the numerical data provided should include all replicates AND the way in which the plotted mean and errors were derived (it should not present only the mean/average values).

REVIEWERS' COMMENTS:

Reviewer #1:

The authors have done a commendable job addressing my concerns. I have no other comments or edits.

Reviewer #2:

My main criticism of the earlier version was that the results were rather obvious - that after doing a lot of work, the authors had come up with main conclusions that we pretty much already knew about. I suggested that the paper could be substantially more interesting, with novel insights, if they did further analyses to see how various species traits - foraging guild, life history traits, major ecological traits, types of taxa etc - might explain some of the variation in responses. Why did some species adjust almost immediately, while others took many generations to respond? Indeed, I even said that if this paper was accepted without these suggested analyses, I would ask to have access to the data set to do those analyses myself. I am pleased that I did not need to do that - that the authors followed the suggestion and that they, in fact, came up with interesting new results. I thank the authors for thanking me for my "generative comments"!

However, I think the paper would be even more impactful if they expanded their Introduction to get readers more excited about and more ready to see the results that lie ahead a page or two later on how foraging guild, or sociality etc affect the loss of fear. I continue to feel that a major aspect of the excitement and insight coming from this paper will be in its analyses of the species traits that affect the changes in antipredator traits. To me, for both knowledge of behaviour and evolution per se, and for conservation/management implications - some of the most interesting results involve insights on which types of species respond more versus less rapidly. The current Introduction only mentions this in its final sentence: "Moreover, we tested how different life history traits may influence the changes in antipredator traits seen in different species". This one sentence does not tell readers about what kinds of traits were examined, what were the hypotheses and why might they be important. I suggest adding a paragraph or two to the Introduction that outline some major hypotheses, their rationale and perhaps even a bit on their implications. For example, why might herbivores change their behaviour more rapidly than carnivores, and why might that be important for understanding human-carnivore-herbivore management? I am, however, not deeply pushy about this suggestion. I think this suggested change will make the paper more exciting for readers, but if the authors prefer not to make that change, I'm OK with that.

COMMENTS FROM THE ACADEMIC EDITOR (lightly edited):

it would be good if they emphasized, even more, why this analysis is important and what practical implications it may have. The paper is very much written as one for insiders, and authors need to learn to reach a larger public and make it easier for the media to pick up what is important about their work. 

Both the abstract and the Introduction both could use some more accessible language. The title doesn't need the word "unraveling" and could state more explicitly what was found rather than what was investigated. The term "Anthropocene" in the title might also help draw attention.

---

## [Editor Report · Decision Letter 3]

17 Aug 2020

Dear Dr Geffroy,

On behalf of my colleagues and the Academic Editor, Frans B M de Waal, I am pleased to inform you that we will be delighted to publish your Research Article in PLOS Biology. 

Early Version

PRESS 

Kind regards,

Alice Musson

Publishing Editor, 

PLOS Biology

on behalf of

Roland Roberts,

Senior Editor

PLOS Biology